# Unsupervised Semantic Segmentation with Self-supervised Object-centric Representations

**Andrii Zadaianchuk[1,3]\*, Matthaeus Kleindessner[2], Yi Zhu[2], Francesco Locatello[2], Thomas Brox[2,4]**
[1]Max-Planck Institute for Intelligent Systems, Tübingen, Germany, [2]Amazon Web Services,
[3]Department of Computer Science, ETH Zürich, [4]University of Freiburg

## Abstract

In this paper, we show that recent advances in self-supervised representation learning enable *unsupervised* object discovery and semantic segmentation with a performance that matches the state of the field on *supervised* semantic segmentation 10 years ago. We propose a methodology based on unsupervised saliency masks and self-supervised feature clustering to kickstart object discovery followed by training a semantic segmentation network on pseudo-labels to bootstrap the system on images with multiple objects. We show that while being conceptually simple our proposed baseline is surprisingly strong. We present results on PASCAL VOC that go far beyond the current state of the art (50.0 mIoU) , and we report for the first time results on MS COCO for the whole set of 81 classes: our method discovers 34 categories with more than 20% IoU, while obtaining an average IoU of 19.6 for all 81 categories.

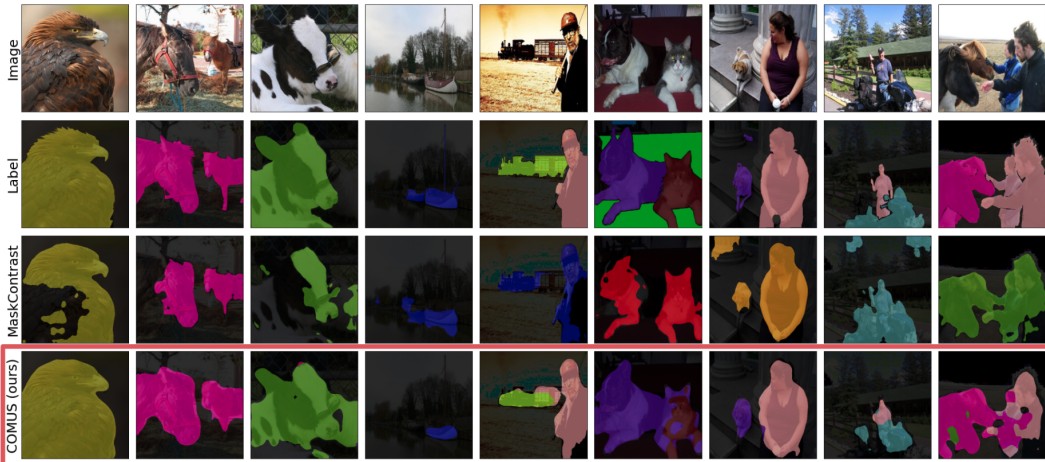

Figure 1: Unsupervised semantic segmentation predictions on PASCAL VOC (Everingham et al., 2012). Our COMUS does not use human annotations to discover objects and their precise localization. In contrast to the prior state-of-the-art method MaskContrast (Van Gansbeke et al., 2021), COMUS yields more precise segmentations, avoids confusion of categories, and is not restricted to only one object category per image.

## 1 Introduction

The large advances in dense semantic labelling in recent years were built on large-scale human-annotated datasets (Everingham et al., 2012; Lin et al., 2014; Cordts et al., 2016). These supervised

---

*Work done during internship at Amazon.

semantic segmentation methods (e.g., Ronneberger et al., 2015; Chen et al., 2018) require costly human annotations and operate only on a restricted set of predefined categories. Weakly-supervised segmentation (Pathak et al., 2015; Wei et al., 2018) and semi-supervised segmentation (Mittal et al., 2019; Zhu et al., 2020) approach the issue of annotation cost by reducing the annotation to only a class label or to a subset of labeled images. However, they are still bound to predefined labels.

In this paper, we follow a recent trend to move away from the external definition of class labels and rather try to identify object categories automatically by letting the patterns in the data speak. This could be achieved by (1) exploiting dataset biases to replace the missing annotation, (2) a way to get the learning process kickstarted based on "good" samples, and (3) a bootstrapping process that iteratively expands the domain of exploitable samples.

A recent method that exploits dataset biases, DINO (Caron et al., 2021), reported promising effects of self-supervised feature learning in conjunction with a visual transformer architecture by exploiting the object-centric bias of ImageNet with a multi-crop strategy. Their paper emphasized particularly the object-centric attention maps on some samples. We found that the attention maps of their DINO approach are not strong enough on a broad enough set of images to kickstart unsupervised semantic segmentation (see Fig. 4), but their learned features within an object region yield clusters of surprisingly high purity and align well with underlying object categories (see Fig. 3).

Thus, we leverage unsupervised saliency maps from DeepUSPS (Nguyen et al., 2019) and BAS-Net (Qin et al., 2019) to localize foreground objects and to extract DINO features from these foreground regions. This already enables unsupervised semantic segmentation on images that show a dominant object category together with an unspectacular background as they are common in PASCAL VOC (Everingham et al., 2012). However, on other datasets, such as MS COCO (Lin et al., 2014), most objects are in context with other objects. Even on PASCAL VOC, there are many images with multiple different object categories.

For extending to more objects, we propose training a regular semantic segmentation network on the obtained pseudo-masks and to further refine this network by self-training it on its own outputs. Our method, dubbed **COMUS** (for **C**lustering **O**bject **M**asks for learning **U**nsupervised **S**egmentation), allows us to segment objects also in multi-object images (see Figure 1), and it allows us for the first time to report unsupervised semantic segmentation results on the full 80 category MS COCO dataset without any human annotations. While there are some hard object categories that are not discovered by our proposed procedure, we obtain good clusters for many of COCO object categories.

Our contributions can be summarized as follows:

1. We propose a strong and simple baseline method (summarized in Figure 2) for unsupervised discovery of object categories and unsupervised semantic segmentation in real-world multi-object image datasets.
2. We show that unsupervised segmentation can reach quality levels comparable to supervised segmentation 10 years ago (Everingham et al., 2012). This demonstrates that unsupervised segmentation is not only an ill-defined academic playground.
3. We perform extensive ablation studies to analyze the importance of the individual components in our proposed pipeline, as well as bottlenecks to identify good directions to further improve the quality of unsupervised object discovery and unsupervised semantic segmentation.

## 2 RELATED WORK

There are several research directions that try to tackle the challenging task of detecting and segmenting objects without any, or with only few, human annotations.

**Unsupervised Semantic Segmentation**   The first line of work (Van Gansbeke et al., 2021; He et al., 2022; Cho et al., 2021; Ji et al., 2019; Hwang et al., 2019; Ouali et al., 2020; Hamilton et al., 2022; Ke et al., 2022) aims to learn dense representations for each pixel in the image and then cluster them (or their aggregation from pixels in the foreground segments) to get each pixel label. While learning semantically meaningful dense representations is an important task itself, clustering them directly to obtain semantic labels seems to be a very challenging task (Ji et al., 2019; Ouali et al., 2020). Thus, usage of additional priors or inductive biases could simplify dense rep-

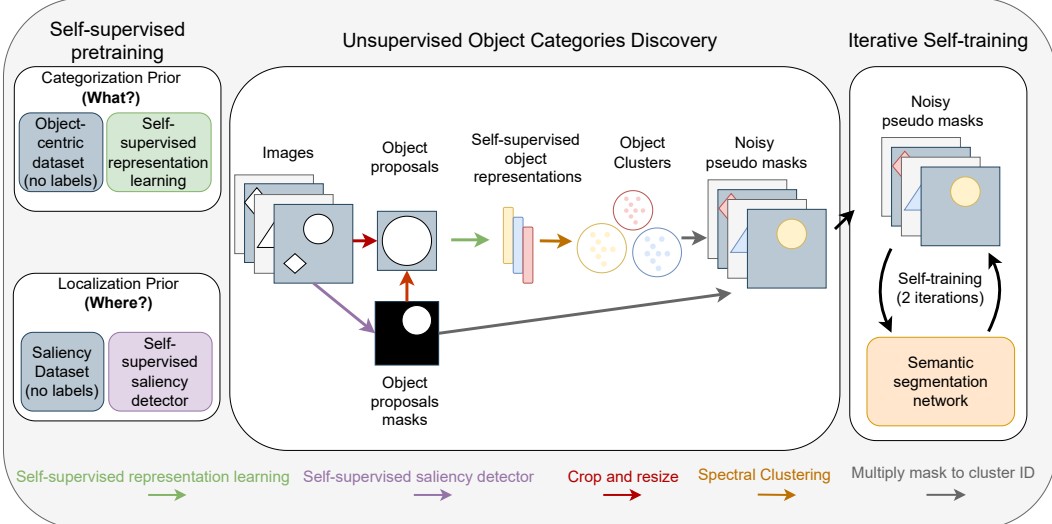

Figure 2: Overview of our self-supervised semantic segmentation framework. First, the self-supervised representation learning network (e.g., DINO (Caron et al., 2021)) and the unsupervised saliency detector (e.g., DeepUSPS (Nguyen et al., 2019)) are trained without manual annotation on object-centric and saliency datasets (e.g., ImageNet (Deng et al., 2009) and MSRA (Cheng et al., 2015)). Next, we use the saliency detector to estimate object proposal masks from the original semantic segmentation dataset. After this, the original images are cropped to the boundaries of object proposal masks and resized. We compute feature vectors within these regions and cluster them with spectral clustering to discover different object categories. We filter the clusters by removing the most uncertain examples. The cluster IDs are combined with the saliency masks to form unsupervised pseudo-masks for self-training of a semantic segmentation network (e.g., DeepLabv3).

resentation learning. PiCIE (Cho et al., 2021) incorporates geometric consistency as an inductive bias to facilitate object category discovery. Recently, STEGO (Hamilton et al., 2022) showed that DINO feature correspondences could be distilled to obtain even stronger bias for category discovery. MaskContrast (Van Gansbeke et al., 2021) uses a more explicit mid-level prior provided by an unsupervised saliency detector to learn dense pixel representations. To obtain semantic labels in an unsupervised way, such representations are averaged over saliency masks and clustered. We show that better representations for each mask could be extracted by using off-the-shelf self-supervised representations from DINO (Caron et al., 2021) encoder. Recently, DeepSpectral (Melas-Kyriazi et al., 2022) proposed to use spectral decomposition of dense DINO features. They suggested over-cluster each image into segments and afterward extracting and clustering DINO representations of such segments while using heuristics to determine the background segment. Those segments represent object parts that could be combined with over-clustering and community detection to improve the quality of pseudo-masks (Ziegler & Asano, 2022). In contrast, we show that starting from object-centric saliency priors discovered on a simpler dataset provides large benefits for discovering object categories (see App. D.1). In contrast to estimating pseudo-masks on the full dataset, using only good quality reliable object proposals for each category combined with iterative self-training on expanding datasets additionally decrease biases over the initial pseudo-masks.

**Unsupervised Object Discovery (UOD)**  UOD is another research direction that also aims to discover object information such as bounding boxes or object masks from images without any human annotations. Recent works on UOD (H'enaff et al., 2022; Melas-Kyriazi et al., 2022; Wang et al., 2022; Simeoni et al., 2021; Vo et al., 2021; Zhang et al., 2020; Vo et al., 2020) showed the potential benefit of using the embeddings of pretrained networks (supervised or self-supervised) for both object localization (to the level of the object's bounding box) and object clustering. First, rOSD (Vo et al., 2021; 2020) showed that supervised features could be used to localize single objects in the image. Next, LOST (Simeoni et al., 2021) proposed a heuristic that relies on self-supervised features to localize the most salient object in the image. In contrast to those methods we consider the challenging task of object segmentation, not only object detection. Finally, Melas-Kyriazi et al.

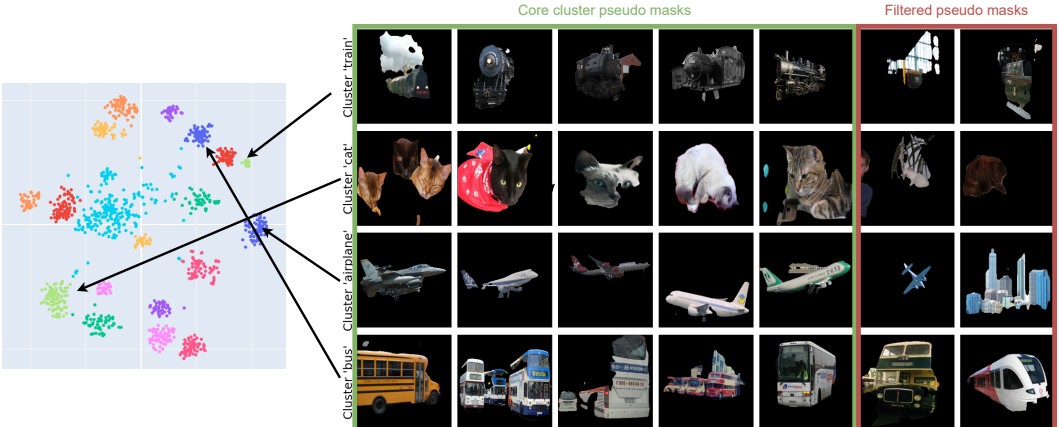

Figure 3: Visualization of unsupervised pseudo-masks on PASCAL VOC *val* set. *(left)* 2D t-SNE projection of object proposal features. Colors correspond to cluster IDs. *(right)* Pseudo-masks from different clusters. The pseudo-masks were randomly sampled for each cluster from both cluster core pseudo-masks (green columns) and filtered pseudo-masks (red columns).

(2022); Wang et al. (2022) propose to do spectral decomposition of dense DINO features and use of sign of Fiedler eigenvector as criteria for object localization mask.

**Unsupervised Object-centric Representation Learning**  Object centric learning assumes that scenes are composed of different objects and aims to learn sets of feature vectors, where each of them binding to one object. Unsupervised methods based on single images (Burgess et al., 2019; Greff et al., 2019; Engelcke et al., 2020; Locatello et al., 2020; Singh et al., 2022a) suffer from single-view ambiguities, which one tries to overcome by exploiting the information in multiple views of a static scene (Chen et al., 2021), in a single view of a dynamic scene (i.e., a video) (Hsieh et al., 2021; Kipf et al., 2021; Singh et al., 2022b) or multiple views of a dynamic scene (Nanbo et al., 2021). In contrast to previous methods and similar to DINOSAUR method (Seitzer et al., 2023), our method exploits unlabeled object-centric datasets to extract object masks and representations.

## 3   SELF-SUPERVISED SEMANTIC SEGMENTATION

### 3.1   INITIAL DISCOVERY OF OBJECT CATEGORIES

Unsupervised decomposition of complex, multi-object scenes into regions that correspond to the present objects categories is hard and largely ill-defined as it is possible to decompose scene with different levels of granularity obtaining several valid decompositions for the same scene (e.g., a person could be considered as one object or additionally decomposed to body parts). However, it is unnecessary to correctly decompose all images of a dataset to kickstart unsupervised object discovery. In this paper, we show that it is sufficient to exploit simple images in a dataset in order to discover *some* objects and their categories. This works particularly well due to intrinsic properties of natural images and photos made by humans. One such property is that the most salient region of the image often corresponds to a single distinct object.

**Self-supervised Object Localization**  Similar to MaskContrast (Van Gansbeke et al., 2021), we propose to retrieve a set of object mask proposals for the images in our dataset by using an unsupervised saliency estimator. In particular, we were using the DeepUSPS (Nguyen et al., 2019) model as an unsupervised saliency estimator. DeepUSPS was trained on MSRA (Cheng et al., 2015) in an unsupervised way exploiting the bias towards simple scenes with often homogeneously textured background of the MSRA dataset, as well as the hard-coded saliency priors of classical (non-learning-based) saliency methods. To further improve the estimator's transfer ability to more complex datasets like PASCAL VOC and MS COCO, we trained another saliency model, Bas-Net (Qin et al., 2019), on the saliency masks generated by DeepUSPS. Some examples of saliency

---

**Algorithm 1:** Object Categories Discovery for Unsupervised Pseudo-Masks Estimation

---

**Given**: $N$ images $x_i$, self-supervised salient regions' segmentation network $L$ with binary threshold $\theta$, self-supervised representation learning method $C$, percentage of proposals to filter $p$.

**Step 1**: Obtain binary object proposal masks $m_i$ by $s_i = L(x_i) > \theta$ and object proposal regions $o_i = \text{crop}(x_i, s_i)$.
**Step 2**: Compute object representations $r_i$ of object proposal regions $o_i$, $r_i = C(o_i)$
**Step 3**: Cluster object proposal representations $r_i$ using spectral clustering to assign cluster ID $t_i$ for each object proposal $o_i$.
**Step 4**: Filter $p$ percents of the most uncertain object proposals for each discovered cluster (proposals with the largest distance to the cluster center in the eigenvalue embedding).
**Step 5**: Combine cluster IDs $t_i$ with object proposal masks $s_i$ to obtain initial pseudo-masks $m_i$.

**Return:** Noisy object segmentation pseudo-masks $m_i$.

---

detector masks are presented in Section 4.3; see Figure 4. In addition, we studied performance of our method with original DeepUSPS masks and with DeepSpectral saliency masks in App. D.

**Self-supervised Representation Learning**  The self-supervised feature learning technique DINO (Caron et al., 2021) exploits the dataset bias of ImageNet, which mostly shows a single object in the center of the image. DINO uses, among other transformations, the multi-crop strategy to link local patterns of the same object instance to its global pattern in the embedding. This leads to a feature representation that tends to have a similar embedding for patterns from the same category.

We start extraction of the object representation by cropping the image to the borders of the saliency mask and resizing the obtained crop to $256 \times 256$ resolution. Next, we feed the object proposal into the Vision Transformer (ViT) (Dosovitskiy et al., 2021) architecture pretrained in a self-supervised way with DINO. The feature vector of the CLS token from the last layer is used as object representation. As CLS token attention values from the last layer of DINO were shown to attend to foreground objects (Caron et al., 2021), the obtained CLS token features are implicitly aggregating object related information from the object proposal.

**Discovery of Semantic Categories**  We cluster the feature vectors obtained per image with spectral clustering (von Luxburg, 2007). Thanks to the saliency masks, most of the feature vectors are based on foreground patterns and disregard the background, i.e., they become object-centric. Even though this is clearly not the case for all images, either because there are salient regions in the background or because the image shows multiple objects from different categories, there are enough good cases for spectral clustering to yield clusters that are dominated by a single object category; see Figure 3. As we show in Table 5, this clustering of features within the saliency masks already yields unsupervised object discovery results beyond the state of the art.

**Filtering of Cluster Samples**  Since neither the salient regions nor DINO features are perfect, we must expect several outliers within the clusters. We tested the simple procedure to filter the most uncertain samples of each cluster and discard them. We measure uncertainty by the distance to the cluster's mean in the spectral embedding of the Laplacian eigenmap (von Luxburg, 2007). In Figure 3 we show that such examples are often failure cases of the saliency detector, such as parts of background that are not related to any category. In addition, we study sensitivity of COMUS algorithm in App. B, showing that COMUS performs comparably well when the percentage of filtered examples varies from $20\%$ to $40\%$. We refer the reader to the Algorithm 1 for the detailed pseudocode of object categories discovery part of the COMUS method.

## 3.2 UNSUPERVISED ITERATIVE SELF-TRAINING WITH NOISY PSEUDO-MASKS

As discussed above, the clustering of feature vectors extracted from within saliency masks makes several assumptions that are only satisfied in some of the samples of a dataset. While this is good enough to get the object discovery process kickstarted, it is important to alleviate these assumptions in order to extend to more samples. In particular, we implicitly relied on the image to show only objects from one category (otherwise the feature vector inside the saliency mask comprises patterns from different categories) and on the correct localization of the object boundaries.

---

**Algorithm 2:** Self-training with Noisy Pseudo-Masks

**Given**: $N$ images $x_i$ with clustering pseudo-masks $m_i$, external $M$ images $x_j$ for self-training

**Step 1**: Train a Teacher network $\theta_t$ (with prediction function $f$) on images with unsupervised pseudo-masks by minimizing the total loss $\mathcal{L}$ for object segmentation:

$$\theta_t^* = \arg\min_{\theta_t} \frac{1}{N} \sum_{j=1}^{N} \mathcal{L}(m_j, f(x_j, \theta_t)).$$

**Step 2**: Generate new pseudo-masks $\widetilde{m}_j$ for all unlabeled images $x_j$ (e.g., images from PASCAL VOC *trainaug* set).
**Step 3**: Train a Student network $\theta_s$ on images and new pseudo-masks $(x_j, \widetilde{m}_j)$:

$$\theta_s^* = \arg\min_{\theta_s} \frac{1}{N+M} \sum_{j=1}^{N+M} \mathcal{L}(\widetilde{m}_j, f(x_j, \theta_s)).$$

**Return:** Semantic segmentation network $\theta_s^*$.

---

To extend also to multi-object images and to improve the localization of object boundaries, we propose using the masks with the assigned cluster IDs as initial pseudo-labels for iterative self-training of a semantic segmentation network. Self-training is originally a semi-supervised learning approach that uses labels to train a teacher model and then trains a student model based on the pseudo-labels generated by the teacher on unlabeled data (Xie et al., 2020). Similar self-training methods were shown to be effective also in semantic segmentation (Chen et al., 2020; Zhu et al., 2020). In this paper, we use the unsupervised pseudo-masks from Sec. 3.1 to train the teacher. In our experiments, we used the network architecture of DeepLabv3 (Chen et al., 2017), but the method applies to all architectures. Since large architectures like DeepLabv3 are typically initialized with an ImageNet pretrained encoder, we also use a pretrained encoder for initialization. However, since we want to stay in the purely unsupervised training regime, we use self-supervised DINO pretraining.

Once the semantic segmentation network is trained on the pseudo-masks, it can predict its own masks. In contrast to the saliency masks, this prediction is not limited to single-object images. Moreover, the training can consolidate the information of the training masks and, thus, yields more accurate object boundaries. Since the masks of the segmentation network are on average better than the initial pseudo-masks, we use them as pseudo-masks for a second iteration of self-training. In addition, if such masks are obtained from unseen images of an extended dataset, the predictions of the segmentation network are not overfitted to the initial pseudo-masks and thus are an even better supervision signal. We refer to the pseudocode in Algorithm 2 for an overview of iterative self-training. In addition, Table 5 in Sec. 4.3 shows that both the initial self-training (Step 1 in Algorithm 2) and the second iteration (Step 3 in Algorithm 2) of self-training improve results.

## 4 EXPERIMENTS

**Evaluation Setting** We tested the proposed approach on two semantic object segmentation datasets, PASCAL VOC (Everingham et al., 2012) and MS COCO (Lin et al., 2014). These benchmarks are classically used for supervised segmentation. In contrast, we used the ground truth segmentation masks only for testing but not for any training. We ran two evaluation settings. For the first, we created as many clusters as there are ground truth classes and did one-to-one Hungarian matching (Kuhn, 1955) between clusters and classes. For the second, we created more clusters than there are ground truth classes and assigned the clusters to classes via majority voting, i.e, for each cluster we chose the class label with most overlap and assigned the cluster to this class. In both cases we used IoU as the cost function for matching and as the final evaluation metric.

Hungarian matching is more strict, as it requires all clusters to match to a ground truth class. Hence, reasonable clusters are often marked as failure with Hungarian matching; for instance, the split of the dominant person class into sitting and standing persons leads to one cluster creating an IoU of 0. This is avoided by majority voting, where clusters are merged to best match the ground truth classes. However, in the limit of more and more clusters, majority voting will trivially lead to a perfect result.

Table 1: Comparison to prior art and iterative improvement via self-training (evaluated by IoU after Hungarian matching) on the PASCAL 2012 *val* set. The results for SwAV and IIC methods are taken from MaskContrast paper. COMUS results are mean $\pm$ standard dev. over 5 runs.

| Method | mIoU |
|---|---|
| Colorization (Zhang et al., 2016) | 4.9 |
| IIC (Ji et al., 2019) | 9.8 |
| SwAV (Caron et al., 2020) | 4.4 |
| MaskContrast (Van Gansbeke et al., 2021) | 35.1 |
| DeepSpectral (Melas-Kyriazi et al., 2022) | $37.2 \pm 3.8$ |
| DINOSAUR (Seitzer et al., 2023) | $37.2 \pm 1.8$ |
| Leopart (Ziegler & Asano, 2022) | 41.7 |
| Pseudo-masks (Iteration 0) | $43.8 \pm 0.1$ |
| COMUS (Iteration 1) | $47.6 \pm 0.4$ |
| COMUS (Iteration 2) | $\mathbf{50.0 \pm 0.4}$ |

Table 2: COMUS performance on PASCAL VOC 2007 *test* (evaluated by IoU after Hungarian matching). The test data was never seen during self-learning or validation.

| | bg | aero | bike | bird | boat | bottle | bus | car | cat | chair | cow | table | dog | horse | mbike | person | plant | sheep | sofa | train | tv | mIoU |
|---|---|---|---|---|---|---|---|---|---|---|---|---|---|---|---|---|---|---|---|---|---|---|
| DeepSpectral | 77.3 | 40.2 | 0.0 | 78.2 | 25.0 | 6.0 | 65.7 | 50.7 | 82.7 | 0.0 | 43.6 | 24.5 | 54.4 | 63.5 | 31.5 | 20.6 | 2.3 | 0.0 | 9.4 | 77.0 | 0.1 | 35.8 |
| COMUS | 84.3 | 38.3 | 30.9 | 51.4 | 47.1 | 39.9 | 66.3 | 54.7 | 67.7 | 0.0 | 60.2 | 21.3 | 54.3 | 57.9 | 62.7 | 45.4 | 9.0 | 72.2 | 13.8 | 81.5 | 43.4 | 47.7 |

When not noted otherwise, we used Hungarian matching in the following tables. We report mean intersection over union (mIoU) as evaluation metric.

**Implementation Details** We used pretrained DINO features with the DINO architecture released in DINO's official GitHub[1]. In particular, we used DINO with patch size 8 that was trained for 800 epochs on ImageNet-1k without labels. For the saliency masks, we used the BasNet weights pretrained on predictions from DeepUSPS released by MaskContrast's official GitHub[2] (see folder `saliency`). All parameters of spectral clustering and self-training are described in App. G.

## 4.1 PASCAL VOC EXPERIMENTS

PASCAL VOC 2012 (Everingham et al., 2012) comprises 21 classes – 20 foreground objects and the background. First, to qualitatively validate that the obtained clusters correspond to true object categories, we visualize the t-SNE embedding (van der Maaten & Hinton, 2008) of DINO representations showing that clusters correspond to different object categories (see Fig. 3). Further, we quantitatively confirmed that saliency masks with assigned cluster ID (pseudo-masks) produce state-of-the-art unsupervised semantic segmentation on PASCAL VOC and outperforms the MaskContrast method that learns dense self-supervised representations; see Table 1 (Iteration 0 row).

For self-training the DeepLabv3 architecture, we initialized the encoder with a ResNet50 pretrained with DINO (self-supervised) on ImageNet and finetuned the whole architecture on the pseudo-masks we computed on the PASCAL 2012 *train* set. This increased the performance from 43.8% mIoU to 47.6% mIoU, see Table 1 (Iteration 1), which supports our consideration of bootstrapping from the original pseudo-masks. In particular, it allows us to segment objects in multi-category images.

Successively, we added one more iteration of self-learning on top of the pseudo-masks on the PASCAL 2012 *trainaug* set. The PASCAL 2012 *trainaug* set (10582 images) is an extension of the original *train* set (1464 images) (Everingham et al., 2012; Hariharan et al., 2011). It was used by previous work on fully-supervised (Chen et al., 2018) and unsupervised (Van Gansbeke et al., 2021) learning. The second iteration of self-training further improves the quality to 50.0% mIoU; see Table 1 (Iteration 2). In particular, it allows us to make multi-category predictions on images from the validation set unseen during self-supervised training (Fig. 1). Accordingly, the method also yields good results on the PASCAL VOC 2007 official test set; see Table 2.

---

[1] https://github.com/facebookresearch/dino
[2] https://github.com/wvangansbeke/Unsupervised-Semantic-Segmentation

Table 3: Unsupervised semantic segmentation before and after self-learning evaluated by mIoU after Hungarian matching on the MS COCO *val* set. As discovered object category we count those categories with an IoU > 20% from all 81 categories. Also, we show IoU for categories that have corresponding cluster (i.e., with IoU larger than zero).

| | all | discovered (with IoU≥ 20%) | | have cluster (with IoU> 0%) | |
|---|---|---|---|---|---|
| | mIoU | number | mIoU | number | mIoU |
| Pseudo-masks | 18.2 | 33 | 36.6 | 73 | 20.2 |
| COMUS | 19.6 | 34 | 40.7 | 60 | 26.5 |

Table 4: Transfer from PASCAL VOC to MS COCO for the 21 PASCAL VOC classes. Training on the simpler PASCAL dataset yields better performance on COCO than learning on COCO itself while both COMUS runs perform better than DeepSpectral.

| | bg | aero | bike | bird | boat | bottle | bus | car | cat | chair | cow | table | dog | horse | mbike | person | plant | sheep | sofa | train | tv | mIoU |
|---|---|---|---|---|---|---|---|---|---|---|---|---|---|---|---|---|---|---|---|---|---|---|
| DeepSpectral | 71.6 | 42.4 | 0.0 | 51.6 | 10.1 | 0.7 | 54.5 | 22.9 | 66.9 | 1.4 | 2.3 | 20.1 | 35.7 | 48.3 | 39.2 | 16.3 | 0.0 | 29.4 | 1.9 | 40.2 | 7.0 | 26.8 |
| COMUS (trained on PASCAL) | 79.5 | 40.7 | 12.4 | 31.9 | 25.7 | 14.0 | 50.6 | 12.1 | 56.1 | 0.0 | 31.0 | 20.1 | 47.6 | 39.6 | 40.6 | 43.5 | 6.8 | 47.6 | 8.0 | 39.7 | 22.8 | 31.9 |
| COMUS (trained on COCO) | 76.5 | 39.9 | 28.2 | 29.7 | 34.3 | 0.1 | 56.8 | 6.8 | 34.9 | 0.7 | 50.2 | 4.4 | 42.1 | 38.7 | 48.4 | 15.1 | 0.0 | 54.4 | 0.0 | 40.4 | 2.6 | 28.8 |

## 4.2 MS COCO EXPERIMENTS

We further evaluated our method on the more challenging COCO dataset (Lin et al., 2014). It focuses on object categories that appear in context to each other and has 80 things categories. We transform the instance segmentation masks to category masks by merging all the masks with the same category together. Our method is able to discover 34 categories with more than 20% IoU. Among those categories, we obtained an average IoU of 40.7%; see Table 3.

Additionally, we studied the transfer properties of COMUS under a distribution shift. To this end, we self-trained our COMUS model on the PASCAL VOC dataset and then tested this model on the same 20 classes on the MS COCO dataset. The results in Table 4 show that the transfer situation between datasets is quite different from supervised learning: training on the PASCAL VOC dataset and testing on COCO yields better results than training in-domain on COCO (see Fig. 10 in Appendix). This is because the PASCAL VOC dataset contains more single object images than MS COCO, which makes self-supervised learning on PASCAL VOC easier. This indicates that the order in which data is presented plays a role for unsupervised segmentation. Starting with datasets that have more single-object bias is advantageous over starting right away with a more complex dataset.

## 4.3 ANALYSIS

**Ablation Study** To isolate the impact of single components of our architecture, we conducted various ablation studies on PASCAL VOC; see Table 5. All proposed components have a positive effect on the result: spectral clustering that additionally computes Laplacian eigenmap before k-means clustering yields better results than k-means clustering (see App. B for detailed analysis on clustering method choice and sensitivity of its parameters); self-training is obviously important to extend to multi-category images; filtering the most distant samples from a cluster followed by the second iteration of self-training on the much larger *trainaug* set gives another strong boost.

**Choice of Categorization Prior** Next, we investigated how COMUS works with different self-supervised representation learning methods. COMUS performs best with ViT based features extractors such as DINO (Caron et al., 2021) and iBOT (Zhou et al., 2022), while its performance is significantly worse for SwAV method (Caron et al., 2020) based on ResNet architecture. We further show that clustering ability of categorization method could be evaluated on ImageNet images clustering where self-supervised methods were originally trained on (see App. C).

**Quality and Impact of Saliency Masks** In Table 7, we compare the quality of the used unsupervised saliency mask detector with other recently proposed detection methods. We report the IoU for the foreground class while using different methods for foreground object segmentation. In particular, we evaluated segmentation masks proposed in LOST (Simeoni et al., 2021) that uses DINO keys correlation between features, and DINOSeg (Simeoni et al., 2021; Caron et al., 2021), which uses the

Table 5: Ablation experiment to identify the effect of individual components of the unsupservised learning process.

| Laplacian Eigenmap | Self-training | Filtering | 2nd self-training | mIoU |
|---|---|---|---|---|
| ✗ | ✗ | ✗ | ✗ | 42.8 |
| ✓ | ✗ | ✗ | ✗ | 43.8 |
| ✓ | ✓ | ✗ | ✗ | 47.1 |
| ✓ | ✓ | ✓ | ✗ | 47.6 |
| ✓ | ✓ | ✓ | ✓ | 50.0 |

Table 6: Comparison of COMUS performance with different feature extractors on PASCAL VOC.

| | mIoU |
|---|---|
| COMUS with SwAV | 28.6 |
| COMUS with iBOT | 43.8 |
| COMUS with DINO | **50.0** |

Figure 4: Visualization of foreground masks obtained with different foreground segmentation methods.

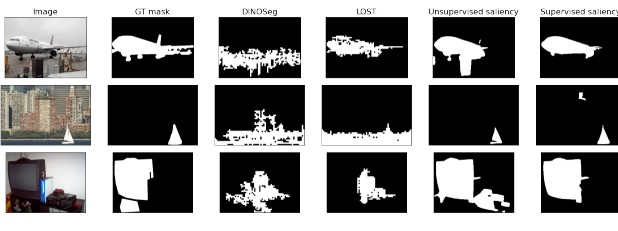

Table 7: Comparison between different class-agnostic foreground segmentation methods.

| | IoU |
|---|---|
| Unsupervised saliency | **51.0** |
| LOST | 34.8 |
| DINOSeg | 24.5 |
| Supervised saliency | 60.5 |

attention weights from the last layer of the DINO network, to construct foreground object segmentation masks (see Figure 4 for examples of predictions by foreground segmentation methods that we consider). The chosen unsupervised saliency based on DeepUSPS and BASNet outperforms both LOST and DINOSeg with a large margin showing the importance of additional localization prior in contrast with relying only on DINO as both categorization and localization prior. In addition, we show how the quality of saliency masks proposal affects COMUS performance in App. D.

**Limitations and Future Work Directions** Although COMUS shows very promising results on the hard tasks of unsupervised object segmentation, there are a number of limitations, as to be expected. First, although we reduced the dependency on the quality of the saliency detector via successive self-training, the approach still fails to segment objects that are rarely marked as salient (see Figure 11 in App. E.2). Second, while the bootstrapping via self-learning can correct some mistakes of the initial discovery stage, it cannot correct all of them and can be itself biased towards self-training data (see App. E.1). Third, we fixed the number of clusters in spectral clustering based on the known number of categories of the dataset. While COMUS works reasonably with larger number of clusters (we refer to App. A for over-clustering experiments), in a fully open data exploration scheme, the optimal number of clusters should be determined automatically.

## 5 CONCLUSION

In this work, we presented a procedure for semantic object segmentation without using any human annotations clearly improving over previous work. As any unsupervised segmentation method requires some biases to be assumed or learned from data, we propose to use object-centric datasets on which localization and categorization priors could be learned in a self-supervised way. We show that combining those priors together with an iterative self-training procedure leads to significant improvements over previous approaches that rely on dense self-supervised representation learning. This combination reveals the hidden potential of object-centric datasets and allows creating a strong baseline for unsupervised segmentation methods by leveraging and combining learned priors.

While research on this task is still in its infancy, our procedure allowed us to tackle a significantly more complex dataset like MS COCO for the first time. Notably, on PASCAL VOC we obtained results that match the best supervised learning results from 2012, before the deep learning era. Hence, the last ten years of research not only have yielded much higher accuracy based on supervised learning, but also allow us to remove all annotation from the learning process.

## ACKNOWLEDGMENTS

We would like to thank Maximilian Seitzer and Yash Sharma for insightful discussions and practical advice. In addition, we thank Vit Musil for PASCAL VOC class icons.

## REPRODUCIBILITY STATEMENT

Algorithm 1 and Algorithm 2 contain pseudocode for both parts of the COMUS method. Appendix G contains detailed information about the COMUS architecture and all hyperparameters used for COMUS training. Appendix H contains information about the datasets used in the paper. The COMUS implementation code is located here: https://github.com/zadaianchuk/comus.

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

# APPENDIX

## A OVER-CLUSTERING

Table 8: Over-clustering results on PASCAL VOC evaluated with mIoU after majority voting. We present the results for 30 clusters, whereas also include the results for 50 clusters for comparison with MaskContrast (Van Gansbeke et al., 2021).

|  | 30 clusters | 50 clusters |
| --- | --- | --- |
| MaskContrast | - | 41.4 |
| COMUS (Iteration 1) | 49.3 | 46.9 |
| COMUS (Iteration 2) | 52.6 | 51.0 |

As the number of discovered clusters could be different from the number of human-defined categories, we ran our process with a larger number of clusters than there are ground truth categories (over-clustering). Each cluster was matched to the ground truth category with the highest IoU, i.e., each category can have multiple clusters being assigned to it. This kind of matching avoids penalization of reasonable subcategories (see Fig. 5). Discovery of subcategories is a strong motivation for using unsupervised methods. Table 8 shows that also under this evaluation protocol we obtain better results. For comparison to MaskContrast, we also included results with 50 clusters, showing that COMUS outperforms MaskContrast in the 50 clusters setting.

Self-training from pseudo-masks with a larger number of categories decreased the performance slightly. Potentially the segmentation network has difficulties learning patterns when several clusters have the same semantic interpretation. In case of fixed saliency masks, this evaluation setting yields better numbers and becomes trivial as the number of clusters becomes very large.

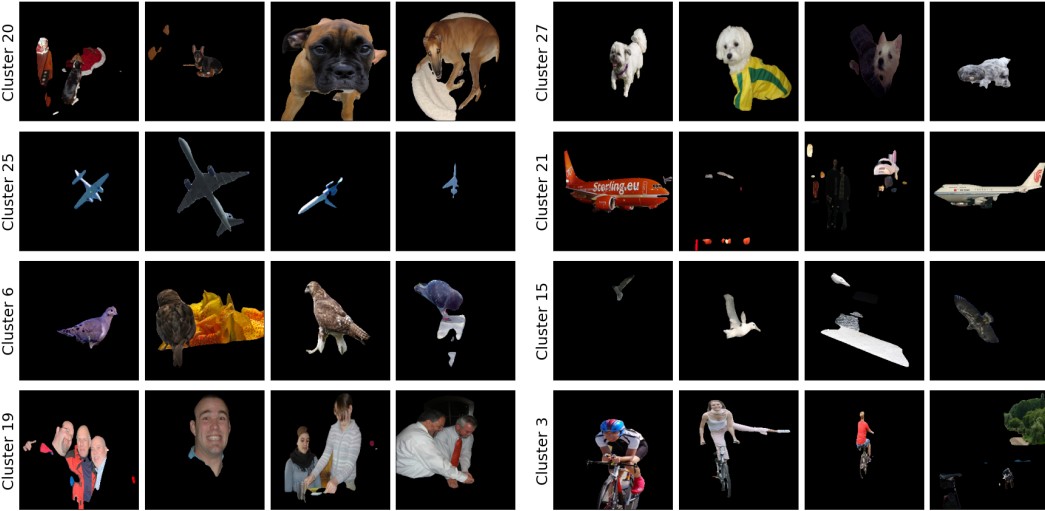

Figure 5: Visualization of discovered subcategories on PASCAL VOC *val* set after clustering of self-supervised representations into 30 clusters. The pseudo-masks were randomly sampled for each cluster. Each row shows two clusters of the same category. The clusters have clear semantic interpretation, such as different dog breeds, flying or staying on land airplanes.

Figure 6: (a) COMUS (Iteration 1) performance with different clustering methods. (b) Effect of "% filtered" on final performance of COMUS. The results are mean ± standard dev. over 5 runs.

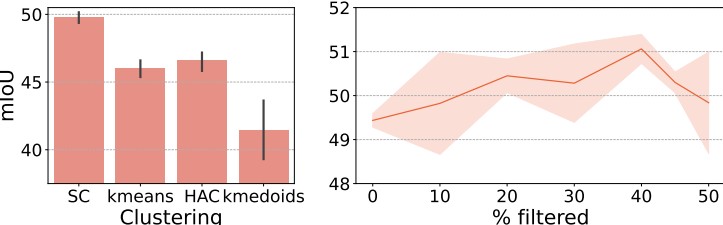

## B    SENSITIVITY OF THE COMUS PARAMETERS

First, we compare COMUS performance with different clustering methods. In Figure 6a we show that SC provides the best supervision signal for COMUS. Additionally, we investigate the sensitivity of COMUS to the choice of % of filtered examples (see Fig. 6b), showing that [20%-40%] of filtered examples leads to comparably good performance. We observe that larger % of filtered examples leads to drop in performance, potentially due to small size of the obtained dataset for training the segmentation network.

Next, we look at the sensitivity of COMUS to the choice of SC parameters. In Table 9 we show that SC is not sensitive to the choice of the number of neighbors and n_init parameters. SC is sensitive to the number of eigenvectors, however, the default value (equal to the number of clusters) shows very good performance. Similar to other works (e.g. Van Gansbeke et al. (2021); Melas-Kyriazi et al. (2022)), we used the same number of clusters as annotated categories (needed for quantitative evaluation).

Finally, we also study the effect of additional iterations of self-training iterations on `trainaug` part of PASCAL VOC dataset. We find that trained for two or three iterations COMUS performs similarly, while for even more self-training iterations we observe slow decrease of the performance (Fig. 7). While we were performing additional self-training iterations on the same dataset, in the future work, it is interesting to study how our method performs in the open-ended regime where new data is available for each iteration.

Figure 7: Number of self-training iterations in COMUS training. The results are mean ± standard dev. over 5 runs.

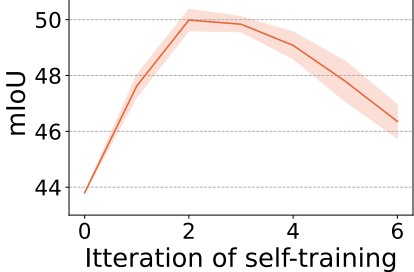

Table 9: Spectral Clustering parameters study, performance after the first iteration.

|         | n_neighbors     | n_components   | n_init         |
|---------|-----------------|----------------|----------------|
| Range   | $[20-50]$       | $[10-40]$      | $[10-60]$      |
| mIoU    | $46.0 \pm 1.3$  | $38.8 \pm 6.2$ | $47.6 \pm 1.0$ |

## C    SELF-SUPERVISED FEATURES QUALITY

As was showed in the original paper (Caron et al., 2021), DINO features are demonstrating excellent performance for k-NN classification (78.3% top-1 ImageNet accuracy), which reveals the quality of the feature space for clustering. In contrast, other self-supervised methods require fine-tuning of last layer (Caron et al., 2020; He et al., 2020) or several last layers (He et al., 2022). We further confirmed (see Table 10) that DINO performs significantly better than SwAV (Caron et al., 2020)

Table 10: Clustering of random subsets of ImageNet classes.

| | Top-1 Accuracy, % | | |
|---|---|---|---|
| | 50 Classes | 100 Classes | 200 Classes |
| SCAN | 76.8 | 68.9 | 58.1 |
| SwAV | $81.6 \pm 0.5$ | $71.5 \pm 0.4$ | $59.2 \pm 0.8$ |
| Supervised | $91.2 \pm 0.9$ | $87.5 \pm 0.3$ | $82.2 \pm 0.4$ |
| DINO | $\mathbf{91.3 \pm 5.1}$ | $\mathbf{88.0 \pm 0.2}$ | $\mathbf{83.1 \pm 0.4}$ |

and SCAN method (Van Gansbeke et al., 2020) (based on MoCo (He et al., 2020) features) for image clustering. For this, similar to SCAN methods (Van Gansbeke et al., 2020), we picked random subsets of ImageNet categories, consisting of 50, 100 and 200 classes. For this experiment, we were using validation images of ImageNet (50 images per category). The results show that DINO features could be used for image clustering with performance comparable with supervised ResNet-50 features.

## D  SALIENCY MASKS QUALITY

In this section, we study how COMUS works with different self-supervised and supervised saliency detectors. Overall, we observe that improving original object proposal masks is important for both category discovery and further iterative self-training. Also, we note that the proposed iterative self-training from filtered pseudo masks is effective for all of the studied choices of saliency detectors.

### D.1  CHOICE OF UNSUPERVISED SALIENCY MASKS DETECTOR

First, we compare COMUS combined with self-supervised BasNet saliency detector (Qin et al., 2019) (i.e., BasNet pretrained with DeepUSPS masks on MSRA-B dataset) with COMUS that is using recently proposed Spectral Decomposition saliency masks from DeepSpectral (Melas-Kyriazi et al., 2022). While original predictions from Spectral Decomposition are performing worse than unsupervised semantic segmentation proposed in DeepSpectral (see the first row of the Table 11), using them in COMUS as objects proposal performs better than DeepSpectral showing the importance of other COMUS components, such as the initial discovery of object categories from object proposals, not from clustered segments (as those are not always object-centric and could cover only parts of objects), filtering of the most uncertain pseudo-masks and several iterations of self-training on previously unseen data. This way, we are able to outperform DeepSpectral semantic segmentation on more than 5 mIoU points, while using only DINO features for both object proposal masks and object representation extraction.

Next, we additionally show how COMUS performs if we use original DeepUSPS masks (Nguyen et al., 2019) as object proposals. As we discussed in the main text, DeepUSPS seems to be less robust than self-supervised BasNet on more complex OOD images from the PASCAL VOC dataset, that was not used in the training (see the first row of the Table 11). Additional self-training iterations are improving the quality of the original object proposals similarly, showing that COMUS can operate with originally lower quality mask proposals in case of successful object categories discovery.

While the original DeepUSPS initialize its backbone weights from supervised pretraining (Nguyen et al., 2019), similar to other areas, it was recently shown that this is not necessary for DeepUSPS good performance (Ryali et al., 2021). They show that DeepUSPS[2] (Ryali et al., 2021) model that does not use any annotations during backbone pretraining still performs comparable to or better than the original DeepUSPS. Thus, similar to the original DeepUSPS saliency masks could be discovered from architecture where no labels were used. As DeepUSPS[2] implementation is not publically available and for better comparison with previous methods in our work, we were using self-supervised BasNet from MaskContrast (Van Gansbeke et al., 2021).

Table 11: Choice of the unsupervised salient object detector. We compared COMUS performance with three different unsupervised saliency masks detectors: self-supervised BasNet model (Qin et al., 2019), original DeepUSPS (Nguyen et al., 2019) and spectral decomposition saliency masks from DeepSpectral (Melas-Kyriazi et al., 2022). All the models are evaluated with by IoU after Hungarian matching on the PASCAL 2012 *val* set.

|  | Self-supervised BasNet | Spectral Decomposition | DeepUSPS |
|---|---|---|---|
| COMUS (Iteration 1) | 47.6 | 43.8 | 45.5 |
| COMUS (Iteration 2) | 50.0 | 45.9 | 47.5 |

### D.2 SUPERVISED SALIENCY MASKS AS LOCALIZATION PRIOR

In addition to using fully self-supervised saliency masks as localization prior, we also consider using BasNet saliency detector trained with supervision on MSRA-B dataset as localization prior. Supervised training of saliency masks leads to even better masks, but drops the property of the method being fully unsupervised. Table 12 shows that supervised saliency masks also improve the final results, as to be expected.

Table 12: Effect of object proposals from supervised saliency detector.

|  | Unsupervised Saliency | Supervised Saliency |
|---|---|---|
| MaskContrast | 35.1 | 38.9 |
| COMUS (Iteration 1) | 47.6 | 50.4 |
| COMUS (Iteration 2) | 50.0 | 52.3 |

## E EXTENDED LIMITATIONS AND FUTURE WORK

### E.1 NUMBER OF SEMANTIC CATEGORIES BIAS

As we are starting our iterative self-training from pseudo-masks restricted to one foreground semantic category per image, it is natural to study how well COMUS can work on more complex images where it is more than one semantic category per image by using discovered categories as an additional signal. Thus, we study COMUS performance in comparison with DeepSpectral (Melas-Kyriazi et al., 2022) on two subsets of PASCAL VOC *val*: the first one is the subset where it is only one foreground semantic category (`subset 1`) and the second one is the subset of images with two or more foreground semantic categories (`subset 2`). We compare several iterations of self-training with DeepSpectral performance using two metrics: first is the standard mIoU showing the overall quality of the predictions. In addition, we compared the precision of recognizing each group. For example for the first subset, it is equal to the percentage of predictions that contain only one foreground semantic category prediction).

For each method, as expected, we observe that the quality of the predictions on the `subset 2` is worse than on the `subset 1`. Iterations of self-training are improving COMUS performance, and allowing reaching better quality (measured by mIoU) on more complex images from `subset 2` than overall DeepSpectral predictions. While improving overall prediction quality, the second iteration of COMUS shows bias towards predicting one foreground semantic category per image. This is potentially due to bias in the PASCAL VOC `trainaug` dataset itself. In contrast, DeepSpectral tends to have the opposite bias toward predicting more than one category per image (i.e., it has lower precision for the first task and higher precision for the second task). This could be due to the image features clustering task that is used by DeepSpectral for segment extraction. Interestingly none of the methods predicts well the number of foreground masks in both subsets. Thus determining the right number of semantic categories for each image is still a challenging problem for unsupervised semantic segmentation and an interesting direction for future work.

Table 13: Number of semantic categories bias. Performance of studied methods on two subsets of PASCALVOC *val* dataset.

| | 1 semantic category | | > 1 semantic category | | overall | |
|---|---|---|---|---|---|---|
| | mIoU | precision, % | mIoU | precision, % | mIoU | precision, % |
| DeepSpectral | 39.9 | 50.9 | 34.5 | 65.7 | 37.1 | 56.2 |
| COMUS (Itteration 1) | 52.0 | 50.5 | 39.9 | 61.8 | 47.6 | 54.6 |
| COMUS (Itteration 2) | 55.2 | 67.7 | 41.2 | 46.5 | 50.0 | 60.1 |

## E.2 FAILURE MODES ANALYSIS

We present illustration of several failure cases in Figure 11. We observe that COMUS still failures to discover some categories, such as `table` category that is treated as a background by saliency method. Also, when an object fills all the background our method fails to fully recover from initial saliency mask assumption, that objects appears only in the foreground. Finally, as our method has only one category for the background, extending COMUS to additionally split backgrounds (e.g., COCO-*Stuff* semantic segmentation masks) is an interesting direction for future work.

## F MORE DETAILED QUANTITATIVE AND QUALITATIVE RESULTS

### F.1 PASCAL VOC

In this subsection, we present additional analysis of COMUS performance on PASCAL VOC dataset. First, in Table 14, we show COMUS performance for each PASCAL VOC category. COMUS performs better than MaskContrast and DeepSpectral on most of the categories. In addition to fully unsupervised semantic segmentation methods, we compare COMUS with recently proposed, weakly-supervised GroupViT method (Xu et al., 2022). GroupViT uses text descriptions as a weak supervision signal to group image regions into progressively larger arbitrary-shaped segments. While it does not require any pixel-level annotations, GroupViT still relies on large annotated datasets containing millions of image-text pairs. On average COMUS performance is worse than GroupViT performance, COMUS performs better on 9 from 21 categories while using no text annotations. In addition, in Figure 12, we visualize COMUS predictions on random images from PASCAL VOC dataset for two stages of COMUS as well as MaskContrast predictions. Finally, for exploring Interactive Demo that visualizes clustering of the whole PASCAL VOC *val* set, visit COMUS project website: https://sites.google.com/view/comuspaper/home.

Table 14: More detailed comparison to prior art and iterative improvement via self-training (evaluated by IoU after Hungarian matching) on the PASCAL 2012 *val* set. Our method results are averaged over 5 runs.

| Method | | | | | | | | | | | | | | | | | | | | | | mIoU |
|---|---|---|---|---|---|---|---|---|---|---|---|---|---|---|---|---|---|---|---|---|---|---|
| *GroupViT (text supervision)* (Xu et al., 2022) | 80.6 | 38.1 | 31.4 | 51.6 | 32.7 | 63.5 | 78.8 | 65.1 | 79.2 | 18.8 | 73.4 | 31.6 | 76.4 | 59.4 | 55.3 | 44.0 | 40.9 | 66.6 | 31.5 | 49.5 | 29.7 | 52.3 |
| MaskContrast (Van Gansbeke et al., 2021) | 84.4 | 39.1 | 20.0 | 59.5 | 34.2 | 38.1 | 57.8 | 60.7 | 46.9 | 0.29 | 0.42 | 3.51 | 28.6 | 39.6 | 54.7 | 23.2 | 0.00 | 40.0 | 14.9 | 54.0 | 37.7 | 35.1 |
| DeepSpectral (Melas-Kyriazi et al., 2022) | 82.1 | 46.1 | 0.0 | 72.6 | 31.9 | 9.1 | 77.3 | 66.1 | 77.5 | 0.1 | 43.4 | 25.9 | 40.6 | 62.6 | 36.9 | 28.0 | 2.5 | 1.1 | 10.8 | 63.9 | 0.0 | 37.1 |
| Pseudo-masks (Iteration 0) | 82.8 | 48.1 | 19.6 | 59.3 | 49.7 | 55.6 | 63.8 | 52.7 | 53.2 | 0.1 | 58.3 | 0.0 | 37.2 | 54.7 | 50.9 | 36.2 | 26.8 | 66.6 | 17.6 | 52.3 | 33.9 | 43.8 ± 0.1 |
| COMUS (Iteration 1) | 85.9 | 51.0 | 21.4 | 49.4 | 52.5 | 61.0 | 71.0 | 61.6 | 65.4 | 0.0 | 47.1 | 16.7 | 59.9 | 48.9 | 56.9 | 49.6 | 33.9 | 58.9 | 16.1 | 56.5 | 36.4 | 47.6 ± 0.4 |
| COMUS | 86.3 | 54.8 | 21.9 | 53.5 | 55.3 | 64.5 | 75.2 | 61.8 | 68.6 | 0.0 | 49.0 | 18.2 | 64.1 | 52.2 | 58.3 | 52.4 | 36.8 | 57.7 | 16.8 | 63.5 | 38.7 | **50.0 ± 0.4** |

### F.2 COCO

In this subsection, we present additional analysis of COMUS performance on COCO dataset. Figure 9 shows the performance of COMUS on COCO for each of 80 COCO categories. We note that COMUS achieves better performance for animal and vehicle categories, as well as salient object categories such as `stop sign` and `traffic light` categories. We also observe that most of the undiscovered categories have small relative object's size (e.g., `spoon`, `remote` and `mouse`). For additional analysis of the connection between relative object's size and COMUS performance,

see Figure 8. In addition, in Figure 13, we visualize COMUS predictions on random images from PASCAL VOC dataset for two stages of COMUS as well as MaskContrast predictions.

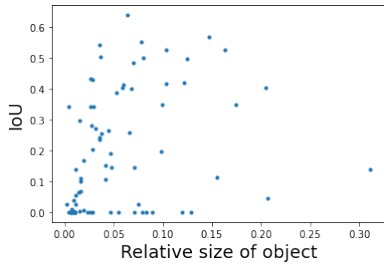

Figure 8: Connection between mean relative size of objects and IoU for each COCO category. The Spearman's rank correlation between relative size and IoU is equal to $0.43$.

## G  IMPLEMENTATION DETAILS

### G.1  SPECTRAL CLUSTERING

We use spectral clustering implementation from sklearn[3]. In particular, an affinity matrix is obtained from the nearest neighbors graph. Number of eigenvectors and number of clusters is the same as number of GT categories. We refer to Table 15 for the parameters of spectral clustering.

Table 15: Spectral clustering parameters for COMUS on PASCAL VOC and MS COCO datasets.

| Hyper-parameter | PASCAL VOC | MS COCO |
| --- | --- | --- |
| Number of clusters | 20 | 80 |
| Number of components | 20 | 80 |
| Affinity | nearest neightbors | nearest neightbors |
| Number of neighbors | 30 | 30 |

### G.2  SELF-TRAINING

During self-training, DeepLabv3 (Chen et al., 2018) with standard cross-entropy loss is chosen to make training set up as comparable to previous research (e.g. to MaskContrast (Van Gansbeke et al., 2021)) as possible. We use CropResize and HorizontalFlip as data augmentation methods. For PASCAL VOC, we perform two iterations of self-training on *train* and *trainaug* sets. For COCO dataset, we perform one iteration of self-training on *train* set. See Table 16 for the parameters of the self-training.

### G.3  COMPUTATIONAL REQUIREMENTS

Similar to other unsupervised segmentation methods (Van Gansbeke et al., 2021; Melas-Kyriazi et al., 2022; Hamilton et al., 2022), the most expensive part of our pipeline, is the training of self-supervised representation learning method. In particular, DINO with Vision Transformers training takes 3 days on two 8-GPU servers and is comparable with other self-supervised representation learning methods. However, learned features could be transferred without further fine-tuning on new natural data images, such as scenes from PASCAL VOC and MS COCO.

The other parts of our method are much faster to train: DeepUSPS could be trained 30 hours of computation time on old four Geforce Titan X (Nguyen et al., 2019), while BasNet could be trained with four GTX 1080ti GPU (with 11GB memory) in around 30 hours (Qin et al., 2019), while the

---

[3]https://scikit-learn.org/stable/modules/generated/sklearn.cluster.SpectralClustering.html

Table 16: Self-training parameters for COMUS on PASCAL VOC and MS COCO datasets.

| Hyper-parameter | PASCAL VOC | MS COCO |
|---|---|---|
| Optimizer | Adam with default settings | Adam with default settings |
| Learning rate | 0.00006 | 0.00006 |
| Batch size | 56 | 56 |
| Input size | 512 | 256 |
| Crop scales | $[0.5, 2]$ | $[0.2, 1.0]$ |
| Number of iterations | 2 | 1 |
| Number of epochs. Iteration 1 | 10 | 1 |
| Number of epochs. Iteration 2 | 5 | - |

inference for 256×256 image only takes 0.040s (25 fps). For more details, we refer the reader to the main papers for these methods.

Finally, the main parts of our method that do require training on new data are object proposals clustering and self-training iterations. Spectral Clustering complexity depends on the sample size. For the full COCO `train` dataset Spectral Clustering with `amg` solver took 20 minutes on 96 core node. Thus, for large datasets, it is recommendable to use its subset for the initial discovery of object categories and then use self-training on the full dataset. For semantic segmentation self-training, we used the standard in supervised semantic segmentation set up for training DeepLabv3 architecture. While all the models could be trained on a single GPU, for convenience we perform all the experiments on one node with 4 NVIDIA T4 GPUs, where 2 iterations of self-training took around one hour.

## H    DATASETS (DIRECTLY OR INDIRECTLY) USED IN THE PAPER

**PASCAL VOC:**    The PASCAL Visual Object Classes (VOC) project (Everingham et al., 2015) provides different datasets / challenges for the years from 2005 to 2012. We apply our proposed method to the datasets from 2012 and 2007, which come with semantic segmentation masks. All datasets and detailed descriptions are available on the PASCAL VOC homepage (http://host.robots.ox.ac.uk/pascal/VOC/index.html).

**MS COCO:**    We also apply our method to the Microsoft (MS) COCO dataset (Lin et al., 2014). The dataset and informations are available on https://cocodataset.org/#home.

**ImageNet:**    For feature extraction, we use vision transformers pretrained with the self-supervised (no labels!) DINO method (Caron et al., 2021) on ImageNet (Deng et al., 2009). The pretrained checkpoint can be found on https://github.com/facebookresearch/dino. Informations about ImageNet are provided on https://image-net.org/.

**MSRA-B:**    For computing saliency mask, we use BasNet (Qin et al., 2019) pretrained on pseudo-labels generated with the unsupervised DeepUSPS (Nguyen et al., 2019) outputs on the MSRA-B dataset (Wang et al., 2017). The pretrained checkpoint can be found on https://github.com/wvangansbeke/Unsupervised-Semantic-Segmentation/tree/main/saliency. The dataset and informations about it are available on https://mmcheng.net/msra10k/.

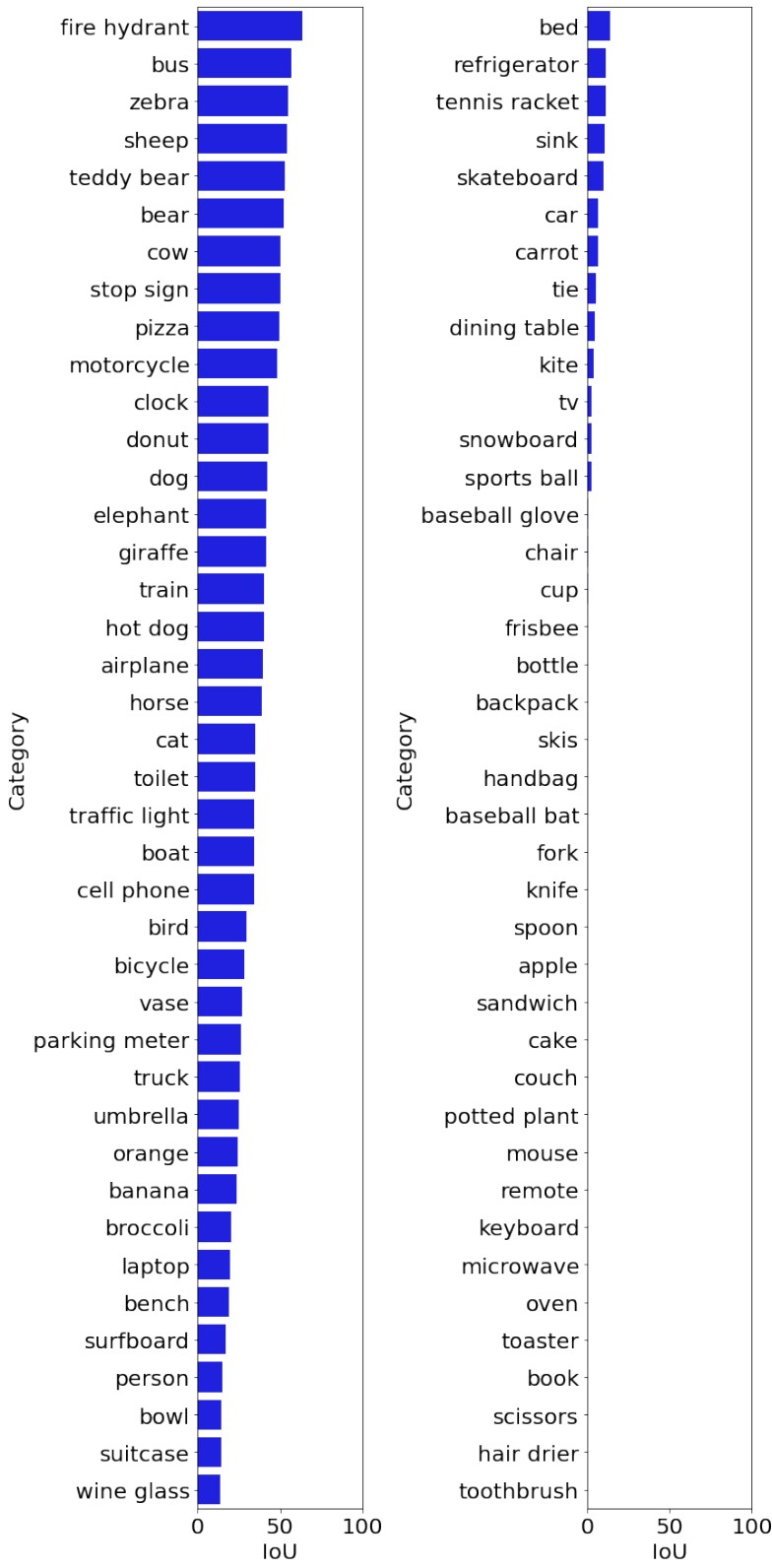

Figure 9: IoU for COCO categories after Hungarian matching of the cluster IDs to ground-truth categories.

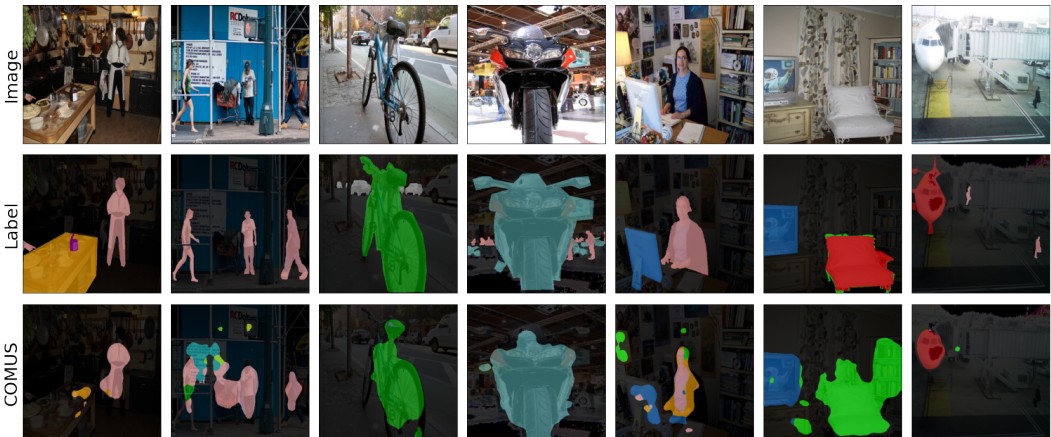

Figure 10: Predictions of the COMUS method trained on PASCAL VOC on COCO *val* set. We notice that the predictions from models trained on PASCAL VOC transfer reasonably well to COCO.

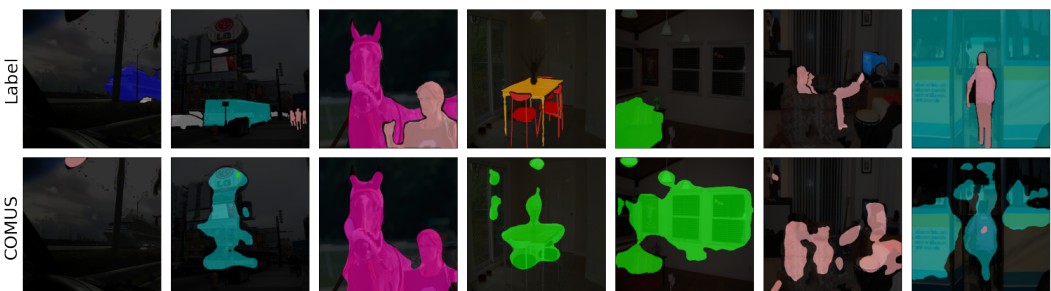

Figure 11: Several failure samples of the COMUS method on PASCAL VOC *val* set. The failures show the limitation and biases of our method, such as bias towards salient objects and misclassifications in multi-object images.

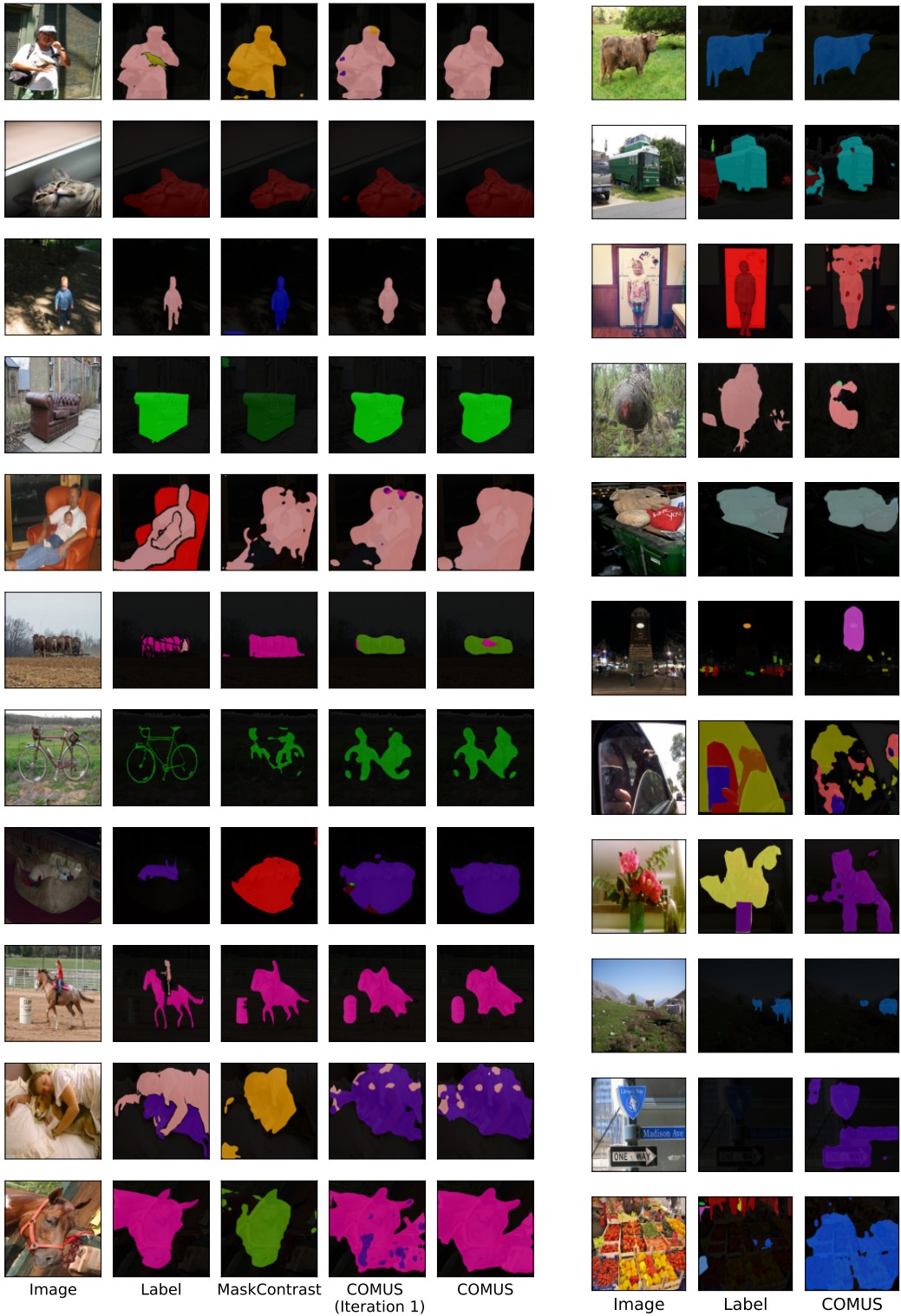

Figure 12: COMUS and MaskContrast predictions on random images from PASCAL VOC *val* set.

Figure 13: COMUS predictions on random images from COCO *val* set.

