# OpenReview forum: "Unsupervised Semantic Segmentation with Self-supervised Object-centric Representations"
_ICLR.cc/2023/Conference — ICLR 2023 notable top 25%_

### Official Review · Reviewer_6jsH · 2022-10-24

**Confidence:** 5
**Correctness:** 3
**Technical Novelty And Significance:** 2
**Empirical Novelty And Significance:** 2
**Recommendation:** 6

**Clarity, Quality, Novelty And Reproducibility:**

Regarding clarity, the paper is well-structured, straightforward, and easy to follow. Enough details are provided that make the paper mostly reproducible. However, due to the reasons outlined above, the proposed idea is not very original and thus the paper lacks in terms of novelty.

**Strength And Weaknesses:**


**Strengths**

The proposed method is simple, yet effective as it achieves over 10 points mIoU above prior work. Results are also shown for the first time on the challenging MS COCO dataset (80 classes).

The method directly builds on pre-trained representations, with optional self-training. As such, and in contrast to some other approaches, it does not require costly self-supervised pre-training at object level (i.e., dense representations) to achieve a pixel-wise task, such as semantic segmentation.


**Weaknesses**

(1) One major issue with the paper is its close resemblance to DeepSpectral (Melas-Kyriazi et al., 2022), although the authors do not discuss this explicitly. DeepSpectral uses a similar pipeline for (unsupervised) semantic segmentation: unsupervised extraction of image regions, clustering for categorizing these regions, and self-training. The most notable difference appears to be the choice of method for the extraction of relevant image regions, that is DeepUSPS for this paper and spectral clustering for DeepSpectral. The rest of the pipeline seems to be quite similar with minor changes (spectral vs k-means clustering, or the number of iterations during self-training). This limits the technical contribution of the paper, despite its strong performance. Therefore, the authors should provide an extensive discussion of the differences to DeepSpectral and highlight their significance.

(2) The approach relies on a saliency detection method (DeepUSPS/BASNet) to get an initial set of object masks. Despite common belief, I would argue that DeepUSPS may not be considered *truly* unsupervised since:
 - it distills an ensemble of *handcrafted* priors
 - it uses *supervised* pretraining (e.g., on ImageNet or Cityscapes according to (Nguyen et al., 2019))

This fact gives an inherent advantage to DeepUSPS vs fully unsupervised approaches for saliency detection. This is also clear from Table 7, where DeepUSPS performs significantly better than approaches such as LOST or DINOSeg which only rely on self-supervised features to estimate saliency. This in turn gives a clear advantage to the object proposal masks used by the rest of the method.

Notably, DeepSpectral also relies on self-supervised features (e.g., DINO) for object proposal masks. Therefore, the question that arises is whether the biggest part of the performance boost of this paper could be simply attributed to the advantage of DeepUSPS over DINO-based saliency. The authors could investigate this further, e.g., by swapping DeepUSPS for the object proposal part of DeepSpectral (namely, spectral decomposition).

(3) In most figures, it is notable that this method still struggles with images containing more than one semantic category. This is due to the initial assumption that there is only a single object per image (i.e., assigning a single cluster ID to the entire salient region in an image). Despite self-training, the model does not seem to fully recover from this assumption, especially in PASCAL VOC where the majority of images do indeed contain a single object, so there is little training signal. This likely can be measured, e.g., by measuring the performance and predicted object count on images containing more than one object

(4) The authors could also consider running experiments on COCO-Stuff, to facilitate comparisons with other state-of-the-art methods, such as STEGO (Hamilton et al., 2022) and PiCIE (Cho et al., 2021).

(5) The authors should also consider citing, discussing, and (where possible) comparing to the following related methods:

[1] Ziegler and Asano, “Self-Supervised Learning of Object Parts for Semantic Segmentation.” (CVPR 2022)

[2] Ke et al. "Unsupervised Hierarchical Semantic Segmentation with Multiview Cosegmentation and Clustering Transformers." (CVPR 2022)

[3] Henaff et al., “Object discovery and representation networks.” (ECCV 2022)


**Summary Of The Paper:**

The paper addresses the problem of unsupervised semantic segmentation from images, which is relatively new in the literature. The methodology is based on saliency detection, clustering of self-supervised features (e.g., DINO) within the salient regions to obtain pseudo-masks, followed by a few iterations of self-training. This method achieves state-of-the-art results on unsupervised semantic segmentation on PASCAL-VOC and MS COCO.

**Summary Of The Review:**

Although this paper achieves state-of-the-art results on the task of unsupervised semantic segmentation by a large margin, these results may be (to a large degree) due to the superior saliency estimation method that is used in this paper in contrast to prior work. The rest of the pipeline is very similar to that of (Melas-Kyriazi et al., 2022). As a result, I find that the novelty of this paper is not significant enough to warrant acceptance to ICLR.

---

> ### Author Response · Authors · 2022-11-18
> **Rebuttal responce (1/2)**
>
> We thank the reviewer for his comments on our work, useful suggestions for additional experiments, and additional related work and will address their concerns in the following.
>
> > The authors should provide an extensive discussion of the differences to DeepSpectral (Melas-Kyriazi et al., 2022) and highlight their significance.
>
> We thank the reviewer for the suggestion. We improved our presentation of the differences with DeepSpectral in the Related Work section and the Appendix by showing and discussing key differences between those methods. In particular, we explained how object-centric object proposals (in contrast to suggested by DeepSpectral segments from over-clustering each image) are important for successful category discovery. This could be because of the smaller transfer gap for DINO feature extraction that was trained on object-centric data. Next, we also explain how filtering the non-reliable proposals and using several iterations of self-training are essential for distilling the right information from proposed pseudo-masks.
> We provide additional experimental results using spectral decomposition masks for the initial object proposals part in COMUS. Usage of those masks in COMUS leads to a large performance difference in comparison to the original proposal from DeepSpectral (see Appendix D.1 in the updated manuscript), showing that it is not only DeepUSPS masks that are essential for the performance of our method. In addition, we included DeepSpectral in our transfer experiments results (cf. Table 4) and evaluated it on the PASCAL VOC test set (cf. Table 2). Finally, we also discuss the different biases of those methods in Appendix E.1 in the updated manuscript.
>
> > DeepUSPS may not be considered truly unsupervised because it distills an ensemble of handcrafted priors
>
> Any unsupervised method would have some handcrafted priors in one or another form for the unsupervised method to work. It is essential that those priors are general enough to be useful. We show that the saliency detector can be used on more complex data out of the scope of where those priors were initially developed and thus illustrating once more that such priors are not restricted to work only on saliency datasets.
>
> > DeepUSPS may not be considered truly unsupervised because DeepUSPS uses supervised pretraining
>
> While the original DeepUSPS uses backbone weights from supervised pretraining, It was shown that they are not necessary for its performance [1]. DeepUSPS^2 does not use any annotations during pretraining and still performs comparable to DeepUSPS, showing that supervised pretraining is not necessary for the good performance of DeepUSPS. As DeepUSPS$^2$ implementation is not publically available and for better comparison with previous methods in our work, we were using self-supervised BasNet from MaskContrast.
>
> > The authors could swap DeepUSPS with the object proposal part of DeepSpectral (namely, spectral decomposition).
>
> Thank you for this suggestion. In the updated version of the paper, we compared (see Appendix D.1 in the updated manuscript) how our method works with several saliency detectors, including spectral decomposition from DeepSpectral that does use only DINO weights for salient object estimation. We showed that while it works worse than our original proposal it improves over DeepSpectral by a large margin. This shows the importance of discovering saliency masks on the simple dataset (difference with DeepUSPS masks) and all the other components of our method (difference with DeepSpectral).
>
> > It is notable that this method still struggles with images containing more than one semantic category
>
> We made an additional analysis of our method performance on the subset of PASCAL VOC images that contain images with only one foreground semantic category (Subset 1) and more than one foreground semantic category (Subset 2) (see Appendix E1 and Table 13). The performance of COMUS increases for both subsets with self-training iterations. Also, we observe that COMUS predictions on more complex Subset 2 outperform DeepSpectral predictions not only on this subset but also on the whole dataset.
> In addition, we study how the number of predictions on each subset corresponds to the GT number of categories. We see that while the first iteration of self-training performs comparable to DeepSpectral (and is biased towards predicting more than one mask), the second iteration of self-training has the opposite bias (COMUS better predicts the number of foreground masks on Subset 1 and predicts worse the number of foreground masks on Subset 2). Interestingly none of the methods predicts well the number of foreground masks in both subsets.
>
> [1] Chaitanya Ryali, David J. Schwab, and Ari S. Morcos. Learning background invariance improves generalization and robustness in self-supervised learning on ImageNet and beyond. In NeurIPS 2021 Workshop on ImageNet.

---

> > ### Author Response · Authors · 2022-11-18
> > **Rebuttal response (2/2)**
> >
> > > The authors could also consider running experiments on COCO-Stuff.
> >
> > While COMUS is not directly applicable to datasets where the background class is additionally categorized, similar to Reviewer m709, we find that extending COMUS to also categorize background classes is an interesting question and a great direction for future work. For example, background class parts of the saliency detector could also be clustered by clustering dense self-supervised DINO features of the background.
> >
> > > The authors should also consider citing, discussing, and (where possible) comparing to additional related methods.
> >
> > We thank the reviewer for additional related work that we were not aware of at the moment of submitting the paper. We have incorporated those methods and several more related works in the related work section and our comparison with other methods.
> >
> > For convenience, we include here new Table 11 from Appendix D of the updated manuscript.
> >
> > Table 11. Choice of the unsupervised salient object detector.
> >
> > |                           | **Self-supervised BasNet** | **Spectral Decomposition** | **DeepUSPS** |
> > |:------------------------------:|:--------------------------:|:--------------------------:|:------------:|
> > | **Pseudo-masks (Iteration 0)** | 41.8                       | 35.4                       | 39.0         |
> > | **COMUS (Iteration 1)**      | 45.2                       | 40.5                       | 42.9         |
> > | **COMUS (Iteration 2)**      | 47.3                       | 42.4                       | 44.4         |

---

### Official Review · Reviewer_hDe9 · 2022-10-26

**Confidence:** 5
**Correctness:** 4
**Technical Novelty And Significance:** 3
**Empirical Novelty And Significance:** 4
**Recommendation:** 8

**Clarity, Quality, Novelty And Reproducibility:**

This work is of high quality and reads well.
The technical novelty of individual steps of the pipeline is not great, but their composition as a whole is.
Although the pipeline is rather complex, there is sufficient detail in the paper on the implementation and the training methodology.

**Strength And Weaknesses:**

I like this work: it is clearly presented, the pipeline is carefully designed in that it leverages some of the latest advances in unsupervised learning, and achieves a substantial improvement of the segmentation accuracy over previous methods in this problem domain.
The experiments also extend to COCO and include some analysis of different design choices for the feature extractor and the saliency detection method. This is very nice.

On the downside, this multi-stage pipeline may look a bit involved, as it requires pre-training of feature embeddings and of a saliency detection network, clustering and multiple rounds of self-training. It would be great if the paper provided more detail on the computational requirements of these individual steps, since such considerations (model complexity) are often part of the game.

Table 2,4 lack some reference. Why not evaluate MaskContrast or DeepSpectral on VOC/COCO for a comparison? Pre-trained models are either publicly available or could be surely requested from the respective authors.

What is the strategy for model selection? (validation data and metric)
What is the effect of more self-training rounds? Would test-time augmentation help (or already used)?

Some analysis of failure modes would be helpful both for understanding the inherent biases in the approach and as inspiration for future work.

(Apologies if I overlooked any answers to these questions — I would be grateful for the pointers.)

**Summary Of The Paper:**

This work develops a pipeline for unsupervised semantic segmentation. The distinguishing feature of this approach is its generalization from training on binary classification examples to multiple classes. This is achieved by clustering feature embeddings extracted with the help of unsupervised saliency detection methods and a self-training formulation of multi-class model training.

**Summary Of The Review:**

Unsupervised semantic segmentation is an emerging problem domain within the realm of unsupervised learning. This paper makes a notable leap and I can see its potential in inspiring a large following.

---

> ### Author Response · Authors · 2022-11-18
> **Rebuttal response**
>
> Dear Reviewer,
> We thank you for your comments, questions, and suggestions for our work and address them in the following.
>
> > It would be great if the paper provided more detail on the computational requirements of these
> individual steps since such considerations (model complexity) are often part of the game.
>
> We have added details on the computational requirements in App. G.3. The most expensive computation, similar to Leopard, DeepSpectral, and MaskContrast is the training of self-supervised features. While this step is expensive, the training of the feature extractor usually needs to be done only on one dataset, and then it can be used to other datasets without retraining.
>
> > Table 2,4 lack some reference. Why not evaluate MaskContrast or DeepSpectral on VOC/COCO for a comparison?
>
> We thank the reviewer for the suggestion. We trained and evaluated DeepSpectral using their official implementation and added a comparison with their method on both the PASCAL VOC test (Table 2) and COCO with VOC classes (Table 4). Furthermore, we also included a detailed per-class DeepSpectral evaluation in Appendix F.1 (Table 14).
>
> > What is the strategy for model selection?
>
> While we tried to tune as few hyperparameters as possible, we were using validation data mask quality to determine optimal settings. While the choice of hyperparameters is an open problem for Unsupervised and Weakly supervised approaches (see ECCV 2020 WSL tutorial: 5. Evaluating weakly-supervised methods[1]) and requires additional validation data, we tried to follow best practice in the field by
> -  Studying the sensitivity of COMUS to its parameters (see Appendix B).
> -  Using default parameters for classical parts of our method like Spectral Clustering.
> -  Reporting the results on the unseen additional test set and studying transferability to the PASCAL subset of COCO.
>
> > What is the effect of more self-training rounds?
>
> We made additional experiments to show the performance of COMUS with more iterations on PASCAL VOC training data (Appendix B, Figure 7). We observe that COMUS performs comparably for 2-3 self-training iterations, while for even more self-training iterations on the same training data, we see a slow decrease in performance.
>
> > Would test-time augmentation help (or already used)?
>
> In the early stages of testing our method, we looked at the usage of TTA for both saliency mask extraction and self-training iterations and found that it was no significant difference.
>
> > Some analysis of failure modes would be helpful both for understanding the inherent biases in the approach and as inspiration for future work.
>
> We are discussing different failure modes in the Limitations and future work section. In the updated version, we also added the Extended Limitations section (Appendix E, Figure 11), where we discuss different biases and failure modes of our method.
>
> [1] https://www.youtube.com/watch?v=D_dEkeb-fto&list=PLcD_yLvcdUll95mAnBDV0rZKhfClJMZMr

---

### Official Review · Reviewer_7EBx · 2022-10-28

**Confidence:** 4
**Correctness:** 4
**Technical Novelty And Significance:** 3
**Empirical Novelty And Significance:** 3
**Recommendation:** 6

**Clarity, Quality, Novelty And Reproducibility:**

The paper is well structured and clearly written. Novelty is due to the proposed pipeline composed of different steps: self-supervised representation learning, self-supervised saliency detection, unsupervised object discovery and iterative self-training.
Reproducibility is not very clear since the authors do not publish their code neither plan to do it and since the method is composed by many steps, it will be difficult to assure that all parameters are clear in order to reproduce the process.

**Strength And Weaknesses:**

The strengths are as follows:
1. The paper is addressing a very important problem - unsupervised semantic segmentation.
2. The pipeline is composed of straighforward techniques with solid backgrdound.
3. The final pipeline is very well justified.
4. State of the art is updated and complete.
5.  Validation is showing the high performance of the method.
6. Public datasets are used to validate the method.
7. Abblation study is presented.
7. Appendices additionally  give insight on the performance of the method.

Weaknesses:
1 Although the pipeline is composed of technically sound techniques, in its essence the novelty is limited to how to combine these techniques. Still, this weakness is considered not so substantial given the good justification of the pipeline and the final segmentation results.
2. My main concern is that the method is not compared to any state of the art unsupervised semantic segmentation technique. In the paper it is said that the performance of the proposed method is comparable to the supervised semantic segmentation but the paper does not report any state of the art (superivesd neither unsupervised segmentation) method performance.
3. It is highly recommendable to apply the same methodology (datasets, statistical measures, etc.) of other state of the art methods for unsupervised semantic segmentation and compare them.


**Summary Of The Paper:**


This paper addresses the problem of unsupervised semantic segmentation with self-supervised object-centric representations. The authors use object-centric datasets on which localization and categorization priors are learned in a self-supervised way, Combining these priors with an iterative self-training procedure allows to obtain highly performing semantic segmentation framework.
The method has been validated on 2 public datasets (MS COCO and PASCAL VOC). The abblation study shows that the method is able to identify individual components of the unsupervised learning process for semantic segmentation. Final results on PASCAL VOC achieved 47,3 of mIoU and on MS COCO - 40,7.

**Summary Of The Review:**

This paper addresses an important problem - unsupervised semantic segmentation. The paper proposes a well justified and technically sound pipeline composed of straightforward steps. Validation is extensive and final results are looking very promissing. However, unfortunately the authors do not compare their method with any state of the art supervised and unsupervised semantic segmentation method.

---

> ### Author Response · Authors · 2022-11-18
> **Rebuttal response**
>
> We thank the reviewer for his comments on our work and will address their concerns in the following.
>
> > My main concern is that the method is not compared to any state-of-the-art unsupervised semantic segmentation technique.
>
> Please note that we do compare our proposed method COMUS to the state-of-the-art methods MaskContrast and DeepSpectral (cf. Table 1) and in the updated version of our paper also with Leopard (cf. Table 1 in the updated manuscript). In addition, we evaluated DeepSpectral on the PASCAL VOC test (cf. Table 2 in the updated manuscript) and PASCAL/COCO dataset (see Table 4 in the updated manuscript) and show that we perform consistently better on all of them.
> In addition, we compare the performance of COMUS and the novel text-supervised semantic segmentation technique GroupViT [1] (Table 14 in the updated manuscript).
> Please let us know if there are other published methods directly comparable with COMUS for which you would want to see a comparison – we promise to include such comparison in the camera-ready version of our paper.
>
> > It is highly recommendable to apply the same methodology (datasets, statistical measures, etc.) of other state-of-the-art methods for unsupervised semantic segmentation and compare them.
>
> We compare with all the other methods on the same dataset (PASCAL VOC val) and the same common measure (mIoU after Hungarian matching). In addition, we added a comparison to DeepSpectral on (PASCAL VOC classes of COCO val dataset) to study the transfer performance of both methods (cf. Table 4).
>
> > Reproducibility is not very clear since the authors do not publish their code neither plan to do it and since the method is composed by many steps, it will be difficult to assure that all parameters are clear in order to reproduce the process.
>
> We promise to release code upon acceptance of our paper (cf. the reproducibility statement following Section 5), which would allow anyone to fully and easily reproduce our experiments.
>
> [1] Jiarui Xu, Shalini De Mello, Sifei Liu, Wonmin Byeon, Thomas Breuel, Jan Kautz, and Xiaolong Wang. GroupViT: Semantic segmentation emerges from text supervision, CVPR 2022

---

### Official Review · Reviewer_m7o9 · 2022-11-03

**Confidence:** 4
**Correctness:** 3
**Technical Novelty And Significance:** 3
**Empirical Novelty And Significance:** 3
**Recommendation:** 8

**Clarity, Quality, Novelty And Reproducibility:**

Clarity/Quality - The paper is well-written, well-structured and clear. Contributions are clearly stated. The whole story of the method and its development is sound and makes it quite an interesting read. Discussion and references to relevant work are on point.

Novelty - The method presents some novel ideas - especially the approach in which the initial pseudo-labels are extracted using the object-centric bias and the spectral clustering procedure and the attribution of classes to each mask without any annotations - afterward I do not consider them as novel ideas since the teacher-student paradigm has been intensively studied before and same goes for the iterative refinement procedure of pseudo labels - definitely useful but not novel. Nonetheless, this is a good example of a simple but effective procedure.

Reproducibility - The authors offered all the necessary details (pseudocode and implementation details). They also mentioned code release upon acceptance.

**Strength And Weaknesses:**

Strengths:
* The tackled problem is challenging and of great interest to the research community.
* State-of-the-art performance (by a large margin compared to very recent published work) on a well-known benchmark PASCAL VOC.
* The first to report results on a more challenging (more object categories) dataset - MS COCO.
* The paper presentation and experimental analysis is well thought out and executed - if it proves the method's efficiency.
* Good ablation studies.
* The dedicated section with the author's discussion regarding the method's limitations is a plus.

Weaknesses:
* The design decision of using another saliency model BasNet on the saliency masks from DeepUSPS is not argued. What are the implications of not doing this step and just using the initial saliency maps provided by DeepUSPS - why is this distillation procedure needed?

Minor comments:
* Algorithm 1 is not referenced in the paper.
* Table 1 caption - MaskContrast is misspelled.
* Figure 4 is first referenced on Page 4, but you need to scroll down to Page 9 to actually see it.

Question: How can this work be extended (if) to semantic segmentation not just object-centric, to be comparable to methods such as STEGO (referenced in the paper)?

**Summary Of The Paper:**

The paper makes a solid contribution toward unsupervised multi-object segmentation in images. The COMUS method leverages unsupervised saliency detectors to initially estimate object proposal masks (for accurate object localization). It then uses self-supervised feature representation networks for feature extraction from the designated region. These representations are clustered into different categories for different object discovery. The cluster IDs are combined with the saliency masks to form an initial set of pseudo-labels used for object segmentation, not just category discovery.

**Summary Of The Review:**

Overall the paper is good and the contribution is solid. The experiment that was particularly interesting was COMUS under a distribution shift - how well it performs out-of-distribution and the implications of starting with easier datasets and gradually increasing the difficulty. The reviewer has no grounds for rejection.

---

> ### Author Response · Authors · 2022-11-18
> **Rebuttal response**
>
> We thank the reviewer for his comments and questions on our work and address them below.
>
> > The design decision of using another saliency model BasNet on the saliency masks from DeepUSPS is not argued. What are the implications of not doing this step and just using the initial saliency maps provided by DeepUSPS - why is this distillation procedure needed?
>
> We have added more details on this design decision in Appendix D.1 of the updated manuscript (see Table 11). Similar to the MaskContrast, we observe that this distillation procedure helps to increase the robustness of the saliency detector on unseen data (i.e., both methods are trained on more simple MSRA-B dataset and applied on more complex dataset PASCAL VOC). To better show this point, we perform additional experiments using only DeepUSPS original masks (see Table 11, third column). We show that our method could be used with different salient object detectors, including DeepUSPS and DeepSpectral. The performance of COMUS with original DeepUSPS masks is worse due to the lower quality of the predictions on PASCAL VOC data while iterative self-training improves the performance for both salient object detectors.
>
> > Question: How can this work be extended (if) to semantic segmentation not just object-centric, to be comparable to methods such as STEGO (referenced in the paper)?
>
> We think that while DINO features CLS token is perfect for the extraction of the object categories presented in the dataset from object proposals, other background parts could also be clustered by clustering dense self-supervised DINO features of the background. Both saliency detector and background clustering could be combined by requiring minimal correspondence between foreground and background parts of dense self-supervised features.
>
>
> For convenience, we include here new Table 11 from Appendix D of the updated manuscript.
>
> Table 11. Choice of the unsupervised salient object detector.
>
> |                           | **Self-supervised BasNet** | **Spectral Decomposition** | **DeepUSPS** |
> |:------------------------------:|:--------------------------:|:--------------------------:|:------------:|
> | **Pseudo-masks (Iteration 0)** | 41.8                       | 35.4                       | 39.0         |
> | **COMUS (Iteration 1)**      | 45.2                       | 40.5                       | 42.9         |
> | **COMUS (Iteration 2)**      | 47.3                       | 42.4                       | 44.4         |

---

### Author Response · Authors · 2022-11-23
**Brief summary of paper revision**

We thank the reviewers for their time and thorough feedback. Below, we summarize how we updated the paper to address reviewers' feedback.
In particular, we added:
- analysis of COMUS performance with DeepSpectral saliency masks (reviewer 6jsH) and original DeepUSPS saliency masks (reviewer m7o9); **Table 11 in Appendix D.1**
- analysis of COMUS performance and DeepSpectral performance on images with only one background category and with more than one background category (reviewer 6jsH); **Table 13 in Appendix E.1**
- detailed evaluation of DeepSpectral on PASCAL VOC test (**Table 2**) and COCO/PASCAL transfer (**Table 4**), reviewers 7EBx and hDe9)
- more self-training iterations of COMUS (reviewer hDe9); **Table 7 in Appendix B**
- computational requirements for COMUS components (reviewer hDe9), **Appendix G.3**

Additionally, we updated the related work section, adding more detailed comparison with DeepSpectral and several recent related works that we were not aware of.

---

### Decision · Program_Chairs · 2023-01-20

**Decision:**

Accept: notable-top-25%

**Justification For Why Not Higher Score:**

it is a good paper but not strong enough in terms of scores

**Justification For Why Not Lower Score:**

good paper solving important problem (unsupervised segmentation) with significant results

**Metareview: Summary, Strengths And Weaknesses:**

This paper deals with challenging problem of unsupervised multi-object segmentation in images. A distinguishing aspect of the proposed method is its generalization from training on binary classification examples to multiple classes, by clustering feature embeddings extracted with the help of unsupervised saliency detection methods and a self-training formulation of multi-class model training. The paper reported substantial improvement of the segmentation accuracy over SOTA methods, especially the first to report results on a more challenging (more object categories) MS COCO dataset.

**Note From Pc:**

if the above contains the word "oral" or "spotlight" please see: "oral" presentation means -> notable-top-5% and "spotlight" means -> notable-top-25%. As stated in our emails, we are disassociating presentation type from AC recommendations

**Summary Of Ac-Reviewer Meeting:**

NA as 3 of 4 reviewers accept the paper, and the 4th reviewer (score 5) is mainly concerned about comparison, which the authors have provided good responses. The 4th reviewer upgraded score to 6 on 7 Jan 2023.